# Resveratrol Alleviates Diabetic Periodontitis-Induced Alveolar Osteocyte Ferroptosis Possibly via Regulation of SLC7A11/GPX4

**DOI:** 10.3390/nu15092115

**Published:** 2023-04-28

**Authors:** Yue Li, Zhijun Huang, Shuaifei Pan, Yuhui Feng, Haokun He, Shuguang Cheng, Lijing Wang, Liping Wang, Janak Lal Pathak

**Affiliations:** Guangzhou Key Laboratory of Basic and Applied Research of Oral Regenerative Medicine, Guangdong Engineering Research Center of Oral Restoration and Reconstruction, Affiliated Stomatology Hospital of Guangzhou Medical University, Guangzhou 510182, China; 2020218955@stu.gzhmu.edu.cn (Y.L.); huangzhijun@stu.gzhmu.edu.cn (Z.H.); 2021210175@stu.gzhmu.edu.cn (S.P.); fengyuhui@stu.gzhmu.edu.cn (Y.F.); 2022210461@stu.gzhmu.edu.cn (H.H.); 2022211378@stu.gzhmu.edu.cn (S.C.); 2018991265@gzhmu.edu.cn (L.W.)

**Keywords:** diabetic periodontitis, ferroptosis, resveratrol, inflammatory mediators, NF-κB signaling

## Abstract

The mode and mechanism of diabetic periodontitis-induced alveolar-osteocyte death are still unclear. This study aimed to investigate the occurrence of ferroptosis in alveolar osteocytes during diabetic periodontitis and the therapeutic potential of resveratrol to alleviate osteocyte ferroptosis. Diabetic periodontitis was induced in C57/BL6-male mice and treated with or without resveratrol. Periodontitis pathogenicity was analyzed by micro-CT and histology, and alveolar-osteocyte ferroptosis was analyzed by immunohistochemistry. MLOY4 osteocytes were treated with *P. gingivalis*-derived lipopolysaccharide (LPS)+advanced glycosylated end products (AGEs) mimicking diabetic periodontitis condition in vitro, with or without resveratrol or ferrostatin-1 (ferroptosis inhibitor). Osteocyte ferroptosis and expression of inflammatory mediators were analyzed. Diabetic periodontitis aggravated periodontitis pathogenicity and inhibited the expression of GPX4 and SLC7A11 in alveolar osteocytes and resveratrol alleviated these effects. LPS+AGEs triggered osteocyte ferroptosis in vitro as indicated by the downregulated GPX4 and SLC7A11, upregulated malondialdehyde, disrupted mitochondrial morphology, and overexpressed pro-inflammatory mediators IL-1β, TNF-α, SOST, RANKL, and IL-6, and ferrostatin-1 or resveratrol treatment reversed these effects. LPS+AGEs upregulated pIKBα and pNF-κB p65 expression in osteocytes, and resveratrol or ferrostatin-1 reversed this effect. In conclusion, diabetic periodontitis triggers alveolar osteocyte ferroptosis possibly via disruption of the SLC7A11/GPX4 axis, and resveratrol has therapeutic potential to correct this biological event.

## 1. Introduction

Periodontitis is a chronic inflammatory oral disease that affects half of the adult population worldwide and is characterized by the loss of periodontal tissues, including periodontal ligaments, cementum, and alveolar bone, that ultimately causes loss of teeth [1]. Food habits, alcohol consumption, smoking, and failure to maintain dental hygiene alter oral microbiome abundance and diversity. Host immune response to the altered oral microbiome is supposed to trigger periodontitis development [2,3,4]. Periodontitis is also associated with various systemic diseases including, cancer, diabetes, liver diseases, osteoporosis, neuronal diseases, etc. [5,6,7]. Moreover, systemic inflammatory diseases also aggravate periodontitis [8]. Diabetes is the most common systemic disease worldwide and has been reported to aggravate periodontitis [9,10]. Diabetes and periodontitis comorbidity (diabetic periodontitis) is more difficult to treat and causes severe effects on periodontal health [11,12,13]. Diabetes-related serum metabolites, high glucose, and advanced glycosylated end products (AGEs) are the main factors that aggravate periodontal inflammation and oral microbiome dysbiosis. Diabetic periodontitis aggravates alveolar bone resorption, causing multiple teeth loss. However, the mechanism of diabetic periodontitis-induced alveolar bone loss is still unclear.

Bone contains three major types of cells, which are bone-forming osteoblasts, bone-resorbing osteoclasts, and matrix-embedded osteocytes. The homeostasis of these three cells’ function is highly important for healthy bones. Diabetic periodontitis has been reported to stimulate osteoclast formation and activity as well as inhibit osteoblast differentiation and matrix mineralization [14,15,16]. Osteocytes, the most predominant cells in bone, account for about 95% of cells in bone, with a half-life of about 25 years [17]. Osteocytes regulate bone homeostasis via regulating osteoblast and osteoclast survival and activities [18]. Inflammatory conditions have been reported to induce osteocyte cell death and disrupt osteocyte-to-osteoblast/osteoclast communication [19]. Inflammatory mediator-treated osteocytes express a higher level of inflammatory cytokines, including sclerostin (SOST), interleukin (IL)-1β, receptor activator of NF-κB ligand (RANKL), tumor necrosis factor (TNF-α), etc. [20,21]. During periodontitis-related inflammation, osteocyte death, and upregulated expression of RANKL, SOST, IL-1β, and TNF-α disrupts bone homeostasis, causing alveolar bone loss [22]. However, the mode of diabetic periodontitis-induced osteocyte death and its underlying mechanism are still unclear. 

Ferroptosis is an iron-related programmed cell death, which was reported by Dixon et al. in 2012 [23,24]. The ferroptosis cascade pathways are divided into the glutathione peroxidase 4 (GPX4) axis, iron metabolism, and lipid peroxidation [25,26]. A recent study has reported osteoblast ferroptosis during diabetic osteoporosis [27]. However, whether diabetic periodontitis induces alveolar osteocyte ferroptosis contributing to alveolar bone loss is still a mystery. Resveratrol is a plant extract with strong anti-inflammatory effects and has been reported to treat periodontitis and periodontitis-induced alveolar bone loss [28]. Zhang X et al. reported that resveratrol inhibits ferroptosis in mice pancreas β MIN6 cells [29]. However, the effect of resveratrol treatment on diabetic periodontitis-induced osteocyte death has not been investigated yet. Moreover, whether resveratrol can inhibit inflammation-induced osteocyte ferroptosis is also unknown. 

This study aimed to investigate the occurrence of alveolar osteocyte ferroptosis during diabetic periodontitis as well as the rescue effect of resveratrol treatment on diabetic periodontitis-induced osteocyte ferroptosis. A diabetic periodontitis mice model was developed and alveolar osteocyte ferroptosis was analyzed. MLOY4 osteocyte culture was treated with diabetic periodontitis condition in the presence or absence of ferroptosis inhibitor ferrostatin-1 to investigate ferroptosis in vitro. Furthermore, resveratrol treatment was performed during in vitro and in vivo studies to analyze the rescue effect of resveratrol on diabetic periodontitis-induced alveolar osteocyte ferroptosis. 

## 2. Materials and Methods

### 2.1. Establishment of Diabetes Periodontitis Model and Resveratrol Treatment

C57/BL6 (six-week-age male mice, weighing 22~25 g) were purchased from Guangzhou Ruige Biotechnology Co., Ltd. (License No. SYXK (Yue) 2021-0249). Guangdong Huawei Testing Co., Ltd. (Guangzhou, China) approved the experimental ethics (HWT-BG-117). Mice were divided into the control group, the diabetic periodontitis group, and the diabetic periodontitis + resveratrol group. The detailed experimental procedure is depicted in Appendix A.

To establish diabetes, mice were fed a high-sugar diet (48 kcal % fat) from 6 weeks to 14 weeks age. At 10 weeks of age, the mice were injected with Streptozotocin (STZ, 25 mg/kg dissolved in 0.1 mmol cold citric acid buffer, pH 4.6) for five consecutive days. Diabetic periodontitis was induced in diabetic mice by maxillary second molar ligation [30] and 100 μL *P. gingivalis*-derived lipopolysaccharides (LPS, 0.2 mg/mL) twice a week injection from 11 to 13 weeks of age. In the control group, 100 μL of normal saline was injected twice a week. The fasting blood glucose level was measured in mice’s tail vein blood using a Glucometer (Beyotime Technology, Shanghai, China). In the resveratrol treatment group, 100 μL (6.5 μM) resveratrol was injected locally in the periodontitis site for 7 days from 13 weeks of age. At 14 weeks of age (8 weeks after the experiment) mice were sacrificed and the maxilla with surrounding periodontal soft tissues was fixed with paraformaldehyde for 2 days. 

### 2.2. Micro-Computed Tomography

The fixed tissues were scanned (SkyScan1172, Bruker-Micro-CT, Kontich, Belgium) under the X-ray tube operated at 96 kV and 65 μA using a 0.5 mm Al filter with a 7.93 μm pixels resolution. The maxillary second molar bone height was measured by SkyScan Dataviewer software. The cement–enamel junction (CEJ) to the alveolar bone crest (ABC) distance was measured at six sites (mesio-buccal, mid-buccal, disto-buccal, mesio-palatal, mid-palatal, and disto-palatal) using Can software (Bruker micro CT). The starting coronal area, defined by a 2D slice 32 μm below the root furcation with a height of 120 μm for the second maxillary molar, was used to analyzed bone volume per total volume (BV/TV), bone surface per total volume (BS/TV), trabecular thickness (Tb.Th), trabecular number (Tb.N), and trabecular separation (Tb.Sp).

### 2.3. Histology and Hematoxylin and Eosin Staining

Fixed maxillae were decalcified in 10% EDTA (Service Technology, G1105, Wuhan, Hubei, China) at room temperature for 30 days. Demineralized tissues were paraffin-embedded and sectioned into 4 μM thick tissue sections. The tissue sections were stained with hematoxylin and eosin (H&E) staining. Images of stained tissue sections were taken (DM4000B-LED/DFC450, Leica, Wetzlar, Germany). The CEJ-ABC distance was measured in the interproximal regions between the first to second molars and the second to third molars, respectively.

### 2.4. Immunohistochemistry

GPX4, a master regulator of ferroptosis, interrupts lipid peroxidation by converting lipid hydroperoxides into non-toxic lipid alcohols [31]. SLC7A11 is a multi-pass transmembrane protein that mediates the cystine/glutamate antiporter activity in the system x_c_^−^ and regulates GPX4 expression and ferroptosis [32]. Therefore, in this study, we chose to perform the immunohistochemistry of GPX4 and SLC7A11. Antigens were retrieved by heating in 4 μM thick tissue sections. Methanol containing 3% hydrogen peroxide was added for 20 min to the tissue section to block endogenous peroxidase. Briefly, bovine serum albumin (10% BSA) was used to block the tissue sections, and the tissue sections were incubated overnight at 4°C with the primary antibody: anti-SLC7A11 (1:100, Santa Cruz, Dallas, TX, USA) and mouse monoclonal anti-GPX4 (1:100, Santa Cruz, Dallas, TX, USA). After rinsing with PBS three times, sections were incubated with goat anti-mouse IgG (H+L) antibody (1:4000, Abcam, ab205719, Cambridge, UK) at room temperature for 30 min. DAB chromogenic agent kit (Boster) was used to develop color, and the samples were counterstained with hematoxylin. The stained tissue sections were observed under the microscope, and images were taken. The visual fields between the first molar and the second molar were taken. The immunostaining positive osteocytes (%) were counted using the Allred scoring system [33]. 

### 2.5. Osteocyte Culture and Treatment

MLOY4 murine osteocytes (Procell, Wuhan, China) were cultured and passaged in DMEM supplemented with 10% (*v*/*v*) FBS (Gibco, Gaithersburg, MD, USA) and 1% penicillin-streptomycin (Gibco, Gaithersburg, MD, USA). MLOY4 cells were divided into five groups: Control, ferrostatin-1, AGEs+LPS, AGEs+LPS+ferrostatin-1, and AGEs+LPS+resveratrol. Ferroptosis inhibitor ferrostatin-1 (1 μM) was added to the MLOY4 cell culture of the ferrostatin-1 group for 24 h [34]. AGEs are advanced glycation end products commonly used to mimic diabetic conditions in vitro cell cultures [35]. LPS is frequently used to mimic the inflammatory condition of periodontitis in vitro [36]. AGEs (200 μg/mL) (Abcam, ab51995, Cambridge, UK) and LPS (1, 2, or 4 μg/mL) (InvivoGen, San Diego, CA, USA) were added to the MLOY4 cell culture of the AGEs+LPS group for 24 h [34]. Ferrostatin-1 (1 μM) [37], AGEs (200 μg/mL) [38], and LPS (4 μg/mL) were added to the MLOY4 cell culture of the AGEs+LPS+ferrostatin-1 group for 24 h. Resveratrol (3.13, 6.25, 12.5, or 25 μg/mL) (Selleck, Shanghai, China), AGEs (200 μg/mL) and LPS (4 μg/mL) were added to MLOY4 cell culture of AGEs+LPS+resveratrol group for 24 h.

### 2.6. RT-qPCR Analysis

Total RNA was isolated using TriZol reagent (Life Technologies, Carlsbad, CA, USA) and NanoDrop spectrophotometer (NanoDrop2000, Wilmington, DE, USA) to analyze the concentration and quality of RNA. Reverse transcription was performed using the PrimeScript RT Master Mix (Takara Bio, Otsu, Japan), and RT-qPCR analysis was performed using SYBR-Green Real-Time PCR Master Mix (Takara Bio, Otsu, Japan) and a Bio-Rad CFX96 PCR detection system. The primers used for RT-qPCR are listed in Table 1. 

### 2.7. Capillary-Based Immunoassay

Whole-cell lysates were extracted using a total protein extraction kit (#9803, CST, Fall River, MA, USA). Protein separation and detection were performed using an automated capillary electrophoresis system (Simple Western system and Compass software: ProteinSimple) as described previously [39]. Wes Separation Capillary Cartridges for 12–230 KD were used for protein separation according to the manufacturer’s standard instructions. The following antibodies were used: GPX4 (Abcam, Cambridge, UK); SLC7A11 (Beyotime, Shanghai, China); IKB*α*, p-IKB*α*, NF-κB p65, and NF-κB p-p65 (Abmart, Shanghai, China); β-actin (Abcam, Cambridge, UK). Signals were detected with an HRP-conjugated secondary anti-rabbit antibody and were visualized using ProteinSimple software.

### 2.8. Cell Viability Assay

MLOY4 cells (2000 cells/well) were seeded in 96-well plates and cultured under different treatment conditions for 24 h. Resveratrol at different concentrations (6.25–200 μg/mL) was added to the MLOY4 cell culture to determine the 50% inhibition growth (IC_50_) of resveratrol (Selleck, Shanghai, China) [40]. Fresh culture medium and CCK-8 (CCK-8, Beyotime Technology, C0037, Shanghai, China) solution were mixed in the ratio of 9:1, added into the culture, and incubated for 1 h at 37 °C. Cell-free culture medium with CCK-8 solution was the blank control. The absorbance at 450 nm was measured by a NanoDrop spectrophotometer (NanoDrop2000, Wilmington, DE, USA). Quantification of cell viability was normalized to the control group.

### 2.9. Enzyme-Linked Immunosorbent Assay

Enzyme-linked immunosorbent assay (ELISA) was performed to measure the protein expression in the conditioned medium. The protein levels expression of *IL-1β* and *TNF-a* was detected in the pg/mL level by using ELISA Standard Kits (RayBiotech, Atlanta, GA, USA). 

### 2.10. Malon-Dialdehyde Assay

The intracellular Malon-dialdehyde assay (MDA) level was measured in the nM/mg level in cell lysate using the lipid peroxidation MDA Detection Kit (Beyotime, S0131M, Shanghai, China)

### 2.11. Transmission Electron Microscopy

After stimulating MLOY4 cells with different concentration groups for 24 h, the cells were fixed with 2.5% glutaraldehyde. Cells were dehydrated, embedded, sectioned, and stained. Finally, cells were photographed using a transmission electron microscope (Hitachi H-7650, Hitachi High-Technologies Corp, Pleasanton, CA, USA).

### 2.12. mRNA Sequencing

Total RNA was isolated using the RNeasy mini kit (Qiagen, Hilden, Germany). Purified RNA was used as input for sequencing library preparation and indexing using the TruSeq stranded mRNA kit (Illumina, CA, USA) according to the manufacturer’s protocol. The RNA-seq libraries were then sequenced using a NextSeq sequencer for 75 cycles of the sequencing reaction. Libraries generated with cBot were diluted to 10 μM and then sequenced on an Illumina NovaSeq 6000 (Illumina, CA, USA) by Haiyi Biotechnology Co. Ltd., Shenzhen, China. High-quality reads were aligned to the GRCm38 genome using the alignment software HISAT2 [41] (hierarchical indexing for spliced alignment of transcripts), and then reads from the genome were further targeted to gene exon regions by HTseq, the number of reads on each gene pair was counted, and the expression of each gene was calculated using the FPKM method. Differential expression analysis of mRNA was performed using the R package DESeq 2. Differentially expressed genes with |log2(FC)| value > 1.0 and q value < 0.05, considered significantly modulated, were used for further analysis.

### 2.13. Statistical Analysis

All data were analyzed with GraphPad Prism 7.0 statistical software (GraphPad, San Diego, CA, USA). Some of the data used an unpaired, non-parametric Mann–Whitney test or a two-tailed unpaired t-test to determine differences between the two groups. Others used one-way analysis of variance (ANOVA) for multiple groups following Dunnett’s test. Data are presented as mean ± SD, and *p* < 0.05 was considered a significant difference.

## 3. Results

### 3.1. Diabetic Periodontitis Triggers Alveolar Osteocyte Ferroptosis

The fasting blood glucose level in diabetic periodontitis mice was ≥17 mmol/L after 2 weeks of STZ injection (Appendix A). We further analyzed the periodontitis status of diabetic periodontitis mice. Micro-CT images showed a highly evident periodontal bone loss in diabetic periodontitis mice compared with control mice (Figure 1A). The CEJ-ABC distance was 2.6-fold higher in diabetic periodontitis mice compared with control mice (Figure 1B). Periodontal bone parameter analysis also revealed the decreased BV/TV, BS/TB, Tb.Th, and Tb.N, and increased Tb.Sp in diabetic periodontitis mice compared with control mice (Figure 1B). Histopathological analysis of H&E-stained periodontal tissue sections also confirmed the prominent bone loss with a higher degree of gingival inflammation and increased CEJ-ABC distance in diabetic periodontitis mice compared with control mice (Figure 1C). We further analyzed the expression of osteocyte ferroptosis markers *GPX4* and *SLC7A11* by immunohistochemistry of periodontal tissue sections. Osteocyte numbers were remarkably reduced with lower expression of *GPX4* and *SLC7A11* in the alveolar bone of diabetic periodontitis mice compared with those of control mice (Figure 1D). Since downregulation of *GPX4* and *SLC7A11* are associated with ferroptosis, our results indicate the occurrence of alveolar osteocyte ferroptosis during diabetic periodontitis.

### 3.2. Diabetes Periodontitis Condition In Vitro Induces Osteocyte Death and Modulates the Expression of Ferroptotic Markers and Inflammatory Cytokines

Among the different concentrations tested, AGEs (200 μg/mL)+LPS (4 μg/mL) induced osteocyte death at 24 h of culture (Figure 2A). Therefore, we used 200 μg/mL of AGEs+4 μg/mL of LPS to mimic a diabetic periodontitis condition in vitro study. AGEs+LPS treatment for 24 h also upregulated the protein level expression of TNF-α and IL-1β in osteocytes (Figure 2B,C). AGEs+LPS treatment downregulated the mRNA expression of *GPX4* and *SLC7A11* in osteocytes (Figure 2D,E). Interestingly, the mRNA expression of proinflammatory cytokines *SOST*, *IL-6*, *TNF-α,* and *RANKL* was upregulated, and anti-inflammatory cytokines *IL-4* and *IL-10* was downregulated in AGEs+LPS-treated osteocytes compared with the control group (Figure 2F).

### 3.3. Inhibition of Ferroptosis Mitigates Diabetic Periodontitis-Induced Osteocyte Cell Death and Inflammation

Inhibition of ferroptosis by ferrostatin-1 partially rescued AGEs+LPS-induced osteocyte cell death (Figure 3A). Similarly, ferrostatin-1 partially upregulated the AGEs+LPS-inhibited *GPX4* and *SLC7A11* expression in osteocytes (Figure 3B,C). Ferroststin-1 treatment also reversed the AGEs+LPS-induced upregulation of *SOST*, *IL-6*, *TNF-α,* and *RANKL* expression in osteocytes (Figure 3D). Moreover, ferrostatin-1 partially upregulated the AGEs+LPS-inhibited *IL-4* and *IL-10* expression in osteocytes (Figure 3D). These results indicate that diabetic periodontitis condition induces osteocyte ferroptosis and osteocyte-mediated inflammation. 

### 3.4. Resveratrol Prevents Diabetic Periodontitis-Induced Osteocyte Ferroptosis

Resveratrol had been reported to treat periodontitis in animal models [42]. Moreover, resveratrol alleviates ferroptosis during early brain injury [43]. However, whether resveratrol can prevent diabetic periodontitis-induced osteocyte ferroptosis has not been investigated yet. We first investigated the IC_50_ of resveratrol in osteocytes. Resveratrol IC_50_ for osteocytes was 53.96 μg/mL (Figure 4A). We analyzed the effect of resveratrol (3.13, 6.25, 12.5, or 25 μg/mL) on AGEs+LPS-treated osteocytes. Resveratrol at a concentration of 6.25 effectively reversed the AGEs+LPS-induced osteocyte cell death (Figure 4B). However, 25 μg/mL resveratrol failed to rescue AGEs+LPS-induced osteocyte cell death and induced *TNF-α* and *IL-1β* expression in osteocytes (Figure 4C,D). Therefore, we used 6.25 μg/mL resveratrol during further in vitro studies. Both resveratrol and ferrostatin-1 treatment partially rescued AGEs+LPS-inhibited protein level expression of GPX4 and SLC7A11 (Figure 4E,F). Resveratrol also partially rescued AGEs+LPS-inhibited *GPX4* and *SLC7A11* mRNA expression in osteocytes (Figure 4G,H). Resveratrol reversed AGEs+LPS-induced upregulation of *SOST*, *IL-6*, *TNF-α,* and *RANKL* in osteocytes (Figure 4I). Moreover, resveratrol upregulated the AGEs+LPS-inhibited *IL-4* and *IL-10* expression in osteocytes (Figure 4I). Resveratrol restored the AGEs+LPS-damaged mitochondrial morphology in osteocytes (Figure 4J). Moreover, AGEs+LPS upregulated the MDA level in osteocytes and this effect was reversed by ferrostatin-1 or resveratrol treatment (Figure 4K). These effects of resveratrol were in a similar trend to the effect of ferrostatin-1 against AGEs+LPS-mediated effects on osteocytes. These results indicate that resveratrol prevents diabetic periodontitis condition-induced osteocyte ferroptosis in vitro. 

### 3.5. Resveratrol Alleviates Diabetic Periodontitis-Induced Ferroptosis of Alveolar Osteocytes In Vivo

We further analyzed the effect of resveratrol treatment on diabetic periodontitis-induced osteocyte ferroptosis in vivo. Resveratrol treatment mitigated the periodontal tissue damage in diabetic periodontitis mice (Figure 5A). Immunohistochemistry data showed recovery of GPX4 and SLC7A11 expression in osteocytes of resveratrol-treated diabetic periodontitis mice compared with that of diabetic periodontitis mice (Figure 5B). These results further support the results of in vitro studies, i.e., 6.5 μg/mL resveratrol mitigates the diabetic periodontitis condition-induced alveolar osteocyte ferroptosis. 

### 3.6. AGEs+LPS and Resveratrol+AGEs+LPS Treatments Modulate NF-κB Signaling in Osteocytes 

The volcano plot and heat map of mRNA sequencing showed 614 differentially expressed genes (DEGs) in the AGEs+LPS group compared with the control group (Appendix A). Among the DGEs, 399 genes were upregulated, and 215 genes were downregulated. Pathway enrichment analysis revealed that the DEGs were attributed to many signaling pathways including ferroptosis and fatty acid metabolisms (Figure 6A). Ferroptosis related-DEGs are shown in the heat map in Appendix A. Similarly, the volcano plot and heat map of mRNA sequencing showed 256 DEGs in the resveratrol+AGEs+LPS group compared with the AGEs+LPS group (Appendix A). Among the DGEs, 133 genes were upregulated, and 123 genes were downregulated. Pathway enrichment analysis revealed that the DEGs were attributed to many signaling, pathways including ferroptosis and TNF signaling (Figure 6A,B). TNF signaling related-DEGs are shown in the heat map in Appendix A. According to the literature, NF-κB of TNF signaling regulated ferroptosis [44]. AGEs+LPS treatment upregulated p-IKBα/IKBα and NF-κB p-p65/NF-κB p65 expression ratio in osteocytes and this effect was reversed by the treatment of ferrostatin-1 or resveratrol (Figure 6C,D; Appendix A). 

## 4. Discussion

Diabetic periodontitis aggravates alveolar bone destruction and causes teeth loss [45]. Diabetic periodontitis promotes alveolar osteoclast differentiation and inhibits osteoblast viability and differentiation [46,47,48]. Osteocytes, the most abundant and long-lived cells in bone, are also affected by diabetic periodontitis. Diabetic periodontitis induces osteocyte death and upregulates the expression of inflammatory factors in osteocytes disrupting osteoblast and osteoclast functions [49,50]. However, the mode of diabetic periodontitis-induced osteocyte death is still unclear. This study elucidated the occurrence of osteocyte ferroptosis during diabetic periodontitis using in vitro and in vivo studies. Diabetic periodontitis-stimulated ferroptotic osteocytes showed higher expression of inflammatory mediators *SOST*, *RANKL*, *IL-1β*, *TNFα*, and *IL-6*. Moreover, anti-inflammatory plant extract resveratrol alleviated diabetic periodontitis-induced alveolar osteocyte ferroptosis, the expression of inflammatory mediators, and periodontitis pathogenicity. These results indicate that diabetic periodontitis induces alveolar osteocyte ferroptosis and resveratrol treatment exerts an anti-ferroptotic effect on osteocytes, alleviating periodontitis pathogenicity (Figure 7). 

Diabetic periodontitis aggravated alveolar bone loss with a higher CEJ-ABC distance. These results are in accordance with the results from the literature [51]. With regard to alveolar bone loss during diabetic periodontitis, the majority of previous studies were mainly focused on osteoblast and osteoclast formation and activity [52,53,54]. Diabetic periodontitis aggravates alveolar bone destruction also via the catabolic effect on osteoblasts and osteocytes [55,56,57]. The emerging importance of osteocytes in bone homeostasis in recent years has drawn the attention of researchers toward the role of osteocytes in inflammatory diseases-induced bone loss [58,59]. Diabetic periodontitis triggers osteocyte death and a higher expression of inflammatory mediators in osteocytes [60,61]. Reports from the recent literature have shown that diabetes induces ferroptosis of various cell types, including osteoblasts [62,63]. System Xc- prevents ferroptosis via the regulation of levels of cysteine and glutathione. SLC7A11 and GPX4 are the key components of system Xc- and the downregulation of these components triggers ferroptosis [64]. This study showed the reduced expression of SLC7A11 and GPX4 in alveolar osteocytes of diabetic periodontitis mice compared with healthy control. Osteocytes cultured in diabetic periodontitis conditions also showed downregulated SLC7A11 and GPX4 mRNA expression. Inhibition of ferroptosis rescued the diabetic periodontitis condition-induced osteocyte cell death as well as diabetic periodontitis condition-inhibited expression of SLC7A11 and GPX4 in osteocytes. These results indicate the occurrence of alveolar osteocyte ferroptosis during diabetic periodontitis. 

Ferroptotic cells cross-link ROS and inflammation via the release of pro-inflammatory damage-associated molecular patterns such as IL-1β, TNF-α, IFN-γ, etc. [65]. Dying osteocytes release various inflammatory mediators, including SOST, RANKL, IL-1β, and TNF-α, which amplify inflammation in the bone microenvironment [66]. These inflammatory mediators inhibit osteoblast function and promote osteoclast formation and activity that causes bone loss [67]. Diabetic periodontitis condition upregulated the expression of pro-inflammatory markers IL-1β, TNF-α, SOST, RANKL, and IL-6, and downregulated anti-inflammatory markers IL-4 and IL-10. Interestingly, inhibition of ferroptosis reverses this effect of diabetic periodontitis on osteocytes. IL-4 and IL-10 induce osteogenic differentiation of osteoblasts, trigger the anti-inflammatory phenotype of macrophages, and inhibit osteoclastogenesis [68,69]. These results indicate that diabetic periodontitis-induced osteocyte ferroptosis triggers higher expression of bone catabolic factors and diminishes the expression of bone anabolic factors. 

Various anti-inflammatory natural products such as icariin, apigenin, and resveratrol have shown anti-ferroptotic effects in various diseases [70,71,72]. The therapeutic application potential of resveratrol on periodontitis has been widely investigated [73]. However, the anti-ferroptotic effect of resveratrol in alveolar osteocytes during diabetic periodontitis has not been investigated yet. This study found that resveratrol (6.25 μg/mL) efficiently rescued the diabetic periodontitis condition-induced osteocyte ferroptosis. Decreased number of mitochondria with altered morphology is characteristic of ferroptosis [74]. Resveratrol restored the diabetic periodontitis condition-altered mitochondria number and morphology in osteocytes. MDA is the secondary product of lipid peroxidation and upregulates during ferroptosis [75]. In this study, both resveratrol and ferrostatin-1 inhibited the diabetic periodontitis-induced MDA level in osteocytes. Moreover, resveratrol also upregulated the expression of GPX4 and SLC7A11 in osteocytes during diabetic periodontitis conditions in vitro in the same fashion that ferrostatin-1 did. Moreover, resveratrol also mitigated the periodontitis pathogenicity, restored the alveolar osteocyte number and morphology, and upregulated osteocytic expression of GPX4 and SLC7A11 in diabetic periodontitis mice. These data indicate that resveratrol has therapeutic potential to prevent alveolar osteocyte ferroptosis during diabetic periodontitis. 

The anti-inflammatory actions of resveratrol by inhibiting the expression of IL-1β, TNF-α, and IL-6 and upregulating the expression of IL-4 and IL-10 in different cell types under different inflammatory conditions have been widely investigated [76]. In this study, resveratrol reversed the diabetic periodontitis-upregulated expression of pro-inflammatory factors IL-1β, TNF-α, SOST, RANKL, and IL-6 in osteocytes. Interestingly, resveratrol also reversed the diabetic periodontitis-downregulated expression of anti-inflammatory factors IL-4 and IL-10 in osteocytes. A similar effect of ferrostatin-1 was observed on the expression of pro- and anti-inflammatory factors in osteocytes during diabetic periodontitis. RANKL and SOST are the key osteocytic factors released from osteocytes during inflammatory conditions that disrupt bone homeostasis [77]. These results indicate that resveratrol inhibits osteocyte-mediated exacerbation of inflammation during diabetic periodontitis in the bone microenvironment, possibly via exerting an anti-ferroptotic effect. This study revealed that resveratrol can mitigate diabetic periodontitis-induced osteocyte-mediated inflammation via inhibition of IL-1β, TNF-α, SOST, RANKL, and IL-6, and upregulation of IL-4 and IL-10 expression in osteocytes. 

mRNA sequencing and bioinformatics analysis revealed that DGEs in diabetic periodontitis condition-treated osteocytes vs. control osteocytes were mainly attributed to ferroptosis and fatty acid metabolisms. Furthermore, DEGs in the diabetic periodontitis vs. resveratrol+diabetic periodontitis group were mainly attributed to ferroptosis, HIF-1 signaling, and TNF signaling. HIF-1 signaling regulates ferroptosis-enhanced diabetic renal tubular injury [78]. Moreover, HIF-1 signaling also participates in GPX4-mediated ferroptosis [79]. NF-κB of TNF signaling also regulates ferroptosis via modulating system Xc-/GPX4 signaling [80]. NF-κB also regulates the HIF-1 signaling via transcriptional regulation of HIF-1α [81]. Based on these findings, we speculate that resveratrol acts on NF-κB signaling to prevent diabetic periodontitis-induced osteocyte ferroptosis. 

In previous studies regarding resveratrol treatment in the periodontitis animal model, resveratrol is administrated mainly via gavage or intraperitoneal injection [28,82,83]. In this study, we locally injected resveratrol 6.5 μg/mL multiple times in the periodontal region. Although the dose of systematically administrated resveratrol is comparatively high, systemically administrated resveratrol can be cleared from the body via urine and feces, minimizing the high-dose related adverse effects. However, higher dose-dependent side effects of resveratrol, including nausea, flatulence, bowel motions, abdominal discomfort, loose stools, and diarrhea, have been reported in various clinical trials [84]. We found that a higher dose of resveratrol induces osteocyte inflammation in vitro and did not rescue the diabetic periodontitis-induced osteocyte cell death. Therefore, the dose-dependent effect of locally administrated resveratrol in periodontitis pathogenicity and osteocyte functions, as well as local adverse effects, should be thoroughly investigated in future studies. The lack of an only periodontitis and an only diabetes group is the limitation of this study. A future study including these groups will provide a better understanding of the role of resveratrol in alleviating diabetic periodontitis comorbidity-induced alveolar osteocyte ferroptosis.

This study showed higher expression of pIKBα and pNF-κB p65 in diabetic periodontitis condition-treated osteocytes, and this effect was reversed by resveratrol treatment in vitro. However, the exact mechanisms of NF-κB signaling on diabetic periodontitis-induced osteocyte ferroptosis and its mitigation by resveratrol should be further investigated using in vitro and in vivo studies. 

## 5. Conclusions

This study revealed the occurrence of alveolar osteocyte ferroptosis during diabetic periodontitis using in vitro and in vivo studies. Diabetic periodontitis mainly disrupted system Xc- via the downregulation of SLC7A11 and GPX4 to induce osteocyte ferroptosis. Diabetic periodontitis-stimulated ferroptotic osteocytes overexpressed pro-inflammatory factors—SOST, RANKL, IL-1β, TNFα, and IL-6—and reduced the expression of anti-inflammatory cytokines. Resveratrol alleviated diabetic periodontitis-induced periodontal pathogenicity, alveolar bone loss, alveolar osteocyte death, and the expression of pro-inflammatory mediators. Moreover, NF-κB signaling was upregulated in diabetic periodontitis condition-treated osteocytes and resveratrol treatment mitigated this effect. These findings elucidate the fact that diabetic periodontitis triggers alveolar osteocyte ferroptosis via the downregulation of the SLC7A11/GPX4-axis, and resveratrol has therapeutic potential to correct this biological event.

## Figures and Tables

**Figure 1 nutrients-15-02115-f001:**
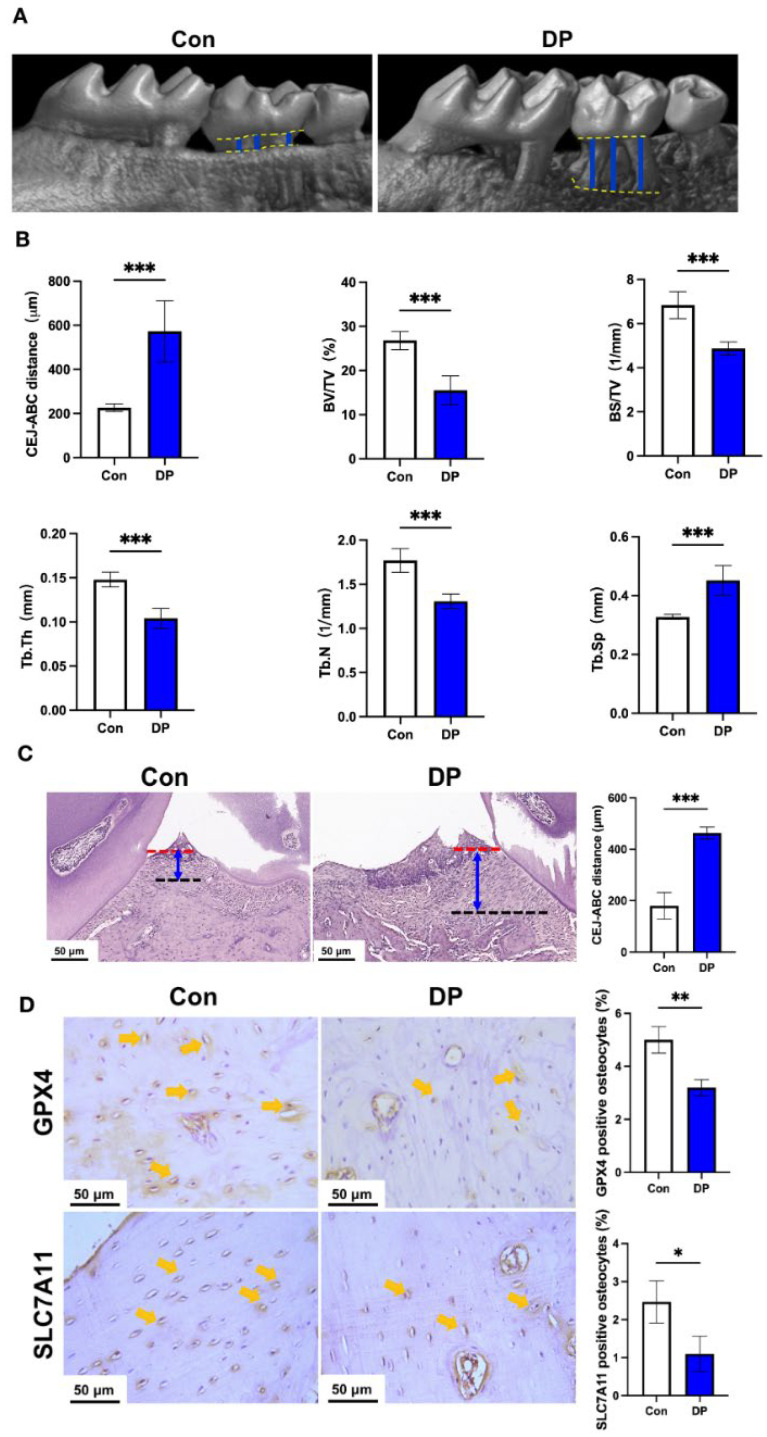
Diabetic periodontitis aggravated periodontitis pathogenicity and reduced the expression of GPX4 and SLC7A11 in alveolar osteocytes. (**A**) Micro-CT 3D reconstruction images of the maxilla of wildtype and diabetic periodontitis mice. (**B**) CEJ-ABC distance, BV/TV, BS/TV, Tb.Th, Tb.N, and Tb.Sp were analyzed from micro-CT images. (**C**) H&E stained histological images of periodontal tissues and the quantitative analysis of CEJ-ABC distance from histological images. (**D**) Representative immunohistochemistry images of GPX4 and SLC7A11 staining and quantification. Data are presented as mean ± SD (*n* = 5). Significant difference compared to the control group, * *p* < 0.05, ** *p* < 0.01, and *** *p* < 0.001. Con, control; DP, diabetes periodontitis; red line, CEJ; black dotted line, ABC; yellow arrow, immunostained osteocyte.

**Figure 2 nutrients-15-02115-f002:**
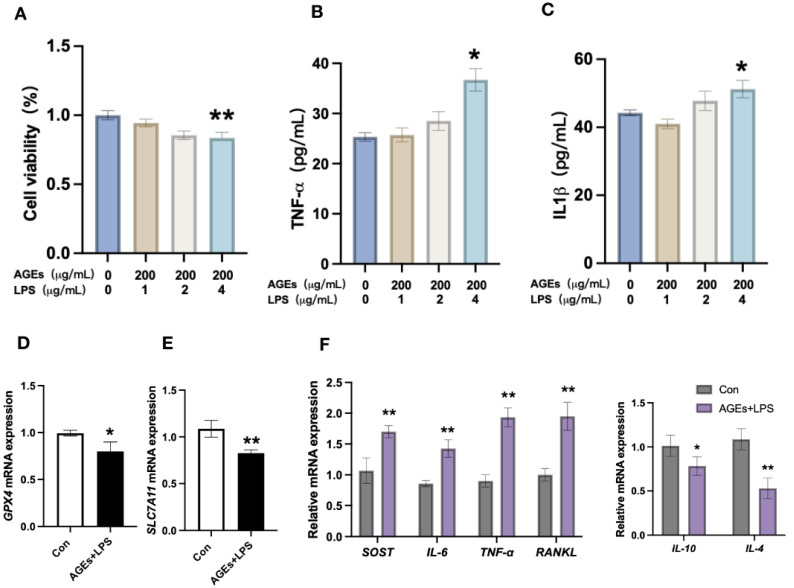
Diabetic periodontitis induced osteocyte death and osteocyte-mediated inflammation. (**A**) Osteocyte viability. Protein level expression of TNF-α (**B**) and IL-1β (**C**) in AGEs+LPS-induced osteocytes analyzed by ELISA. Relative mRNA expression of GPX4 (**D**), SLC7A11 (**E**). (**F**) Relative expression of pro-inflammatory markers SOST, IL-6, TNF-α, and RANKL, and anti-inflammatory markers IL-10 and IL-4. Data are presented as mean ± SD (*n* = 3). Significant difference compared to the control group, * *p* < 0.05 and ** *p* < 0.01. Con, Control; AGEs, advanced glycation end products; LPS, lipopolysaccharide.

**Figure 3 nutrients-15-02115-f003:**
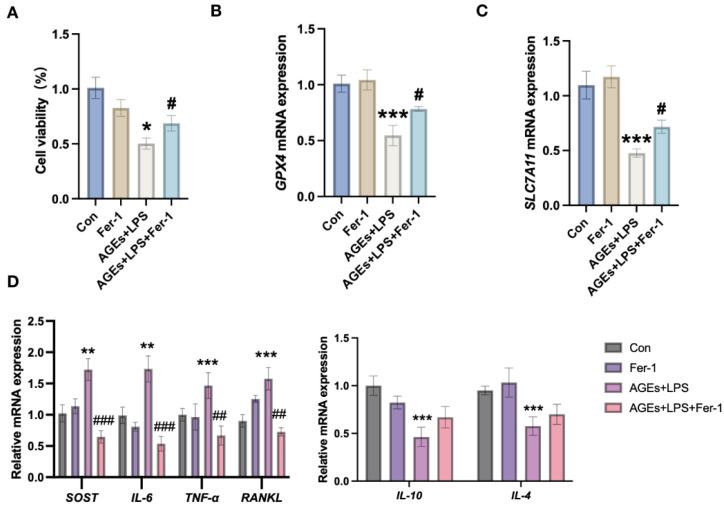
Diabetic periodontitis condition promoted osteocyte ferroptosis and inflammation. (**A**) Osteocyte viability. (**B**,**C**) Relative mRNA expression of ferroptosis markers *SLC7A11* and *GPX4* in osteocytes. (**D**) Relative mRNA expression of pro-inflammatory markers and anti-inflammatory markers in osteocytes. Data are presented as mean ± SD (*n* = 3). Significant difference compared to control and ferrostatin-1 group, * *p* < 0.05, ** *p* < 0.01, and *** *p* < 0.001, and compared to AGEs+LPS group, ^#^ *p* < 0.05, ^##^ *p* < 0.01, and ^###^ *p* < 0.001. Con, control; AGEs, advanced glycation end products; LPS, lipopolysaccharide; Fer-1, ferrostatin-1.

**Figure 4 nutrients-15-02115-f004:**
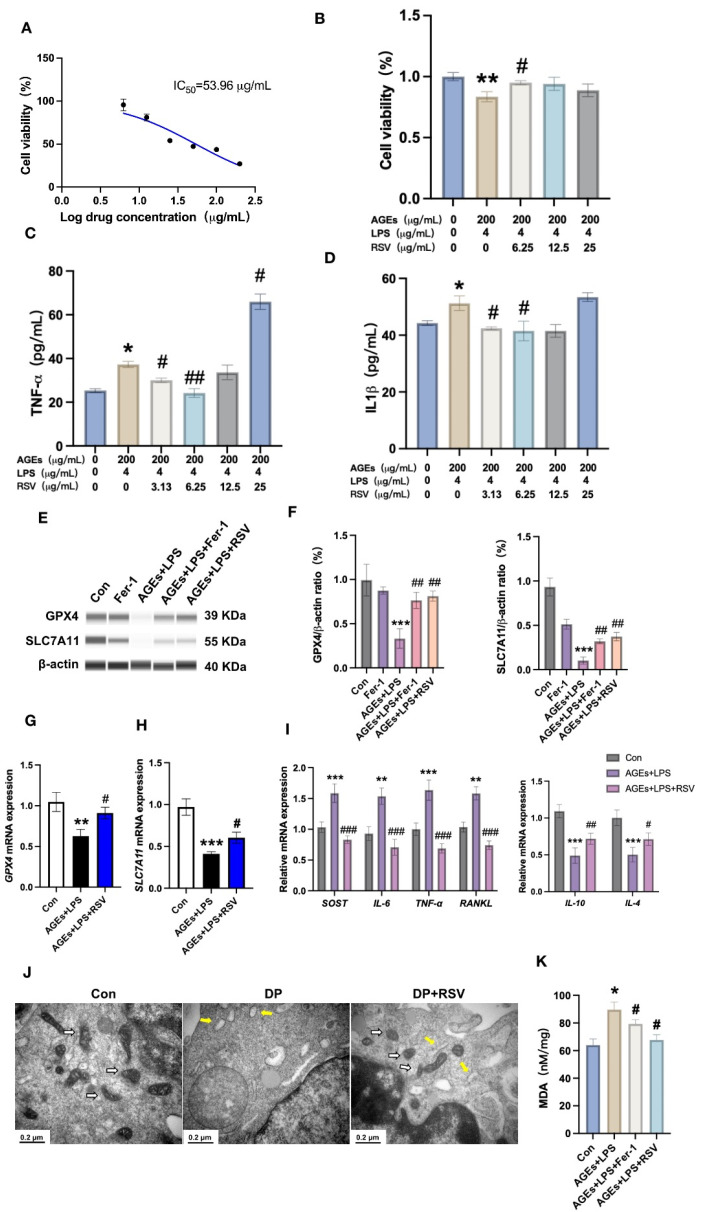
Resveratrol reversed the diabetic periodontitis condition-induced osteocyte ferroptosis in vitro. (**A**) The half maximum inhibitory concentration (IC_50_) of resveratrol (RSV) in osteocytes. (**B**) Osteocyte viability. Protein level expression of TNF-α (**C**) and IL-1β (**D**) in osteocytes analyzed by ELISA. (**E**,**F**) Capillary-based immunoassay analysis and quantification of GPX4 and SLC7A11 in osteocytes. (**G**,**H**) Relative mRNA expression of *GPX4* and *SLC7A11* in osteocytes. (**I**) Relative mRNA expression of pro-inflammatory markers and anti-inflammatory markers in osteocytes. (**J**) TEM images of osteocytes showing mitochondrial damage (white arrow: normal mitochondria; yellow arrow: damaged mitochondria). (**K**) Analysis of MDA level in osteocyte cell lysate. Data are presented as mean ± SD (*n* = 3). Significant difference compared to the control or the first group, * *p* < 0.05, ** *p* < 0.01, and *** *p* < 0.001, and compared to the AGEs+LPS group, ^#^ *p* < 0.05, ^##^ *p* < 0.01, and ^###^ *p* < 0.001. Con, control; AGEs, advanced glycation end products; LPS, lipopolysaccharide; Fer-1,ferrostatin-1; RSV, resveratrol.

**Figure 5 nutrients-15-02115-f005:**
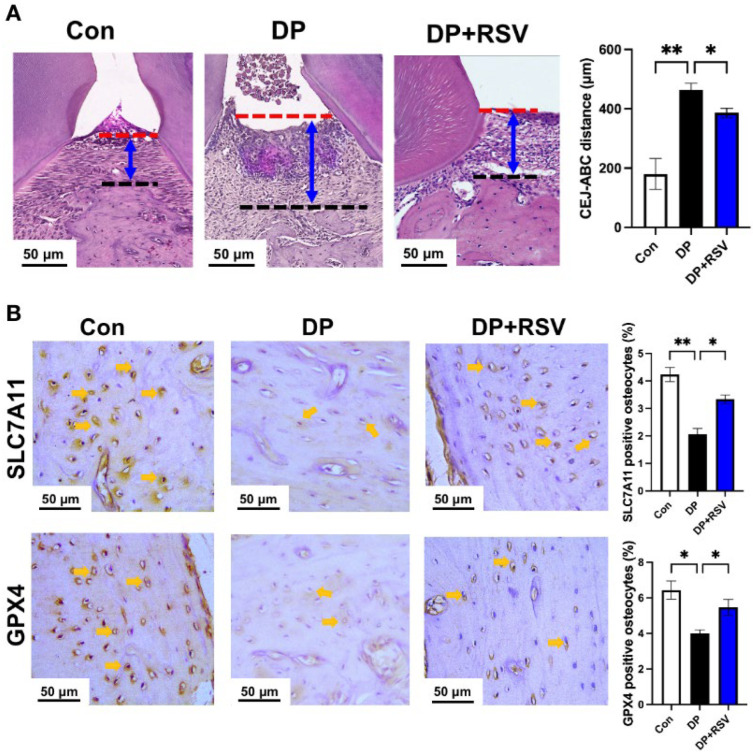
Resveratrol treatment mitigated diabetic periodontitis-induced osteocyte ferroptosis in mice. (**A**) Representative H&E staining images and CEJ-ABC distance analysis. (**B**) Representative immunohistochemistry images of GPX4 and SLC7A11 staining and quantification of GPX4 and SLC7A11 positive osteocytes in the alveolar bone. Data presented as mean ± SD (*n* = 5). The significant difference between the groups, * *p* < 0.05 and ** *p* < 0.01. Con, control; DP, diabetes periodontitis; RSV, resveratrol; red line, CEJ; black dotted line, ABC; yellow arrow, immunostained osteocyte.

**Figure 6 nutrients-15-02115-f006:**
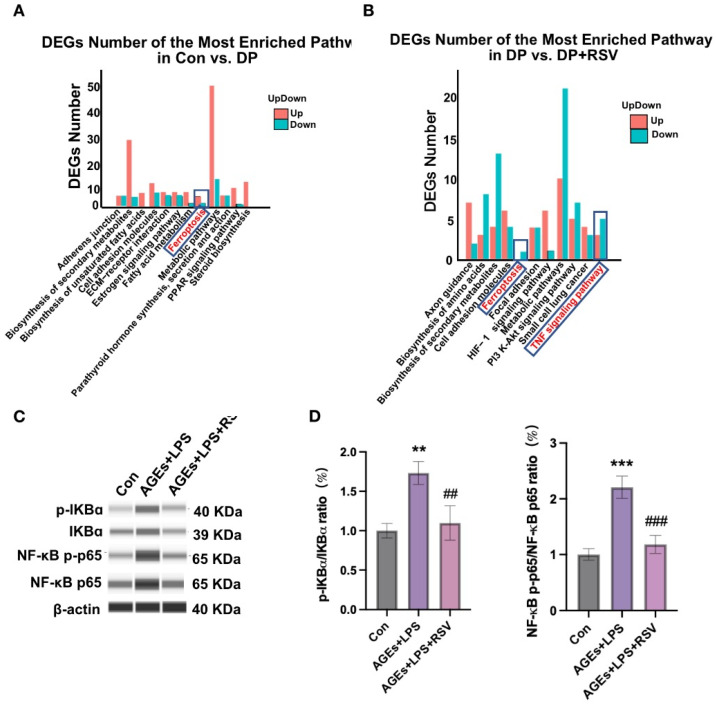
(**A**) pathway enrichment analysis of differential genes in Con vs. DP. (**B**) pathway enrichment analysis of differential genes in DP vs. DP+RSV. (**C**,**D**) Capillary-based immunoassay analysis and quantification of p-IKBα/IKBα and NF-κB p-p65/NF-κB p65 proteins in MLOY4 cells treated under different conditions. Data are presented as mean ± SD (*n* = 3). Significant difference: compared to the control group, ** *p* < 0.01 and *** *p* < 0.001, and compared to the AGEs+LPS group, ^##^ *p* < 0.01 and ^###^ *p* < 0.001. Con, control; AGEs, advanced glycation end products; LPS, lipopolysaccharide; Fer-1, ferrostatin-1; RSV, resveratrol; blue box, cell death related pathway.

**Figure 7 nutrients-15-02115-f007:**
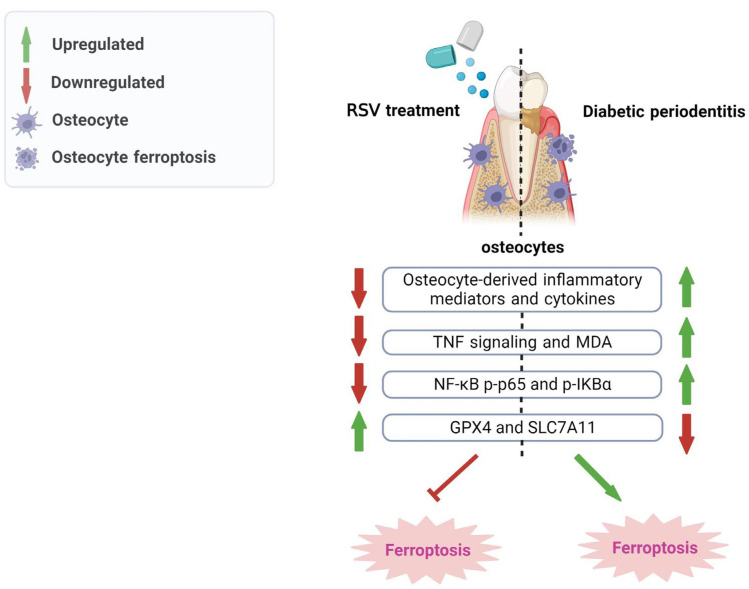
Scheme showing the protective effect of resveratrol against diabetic periodontitis-induced alveolar osteocyte ferroptosis.

**Table 1 nutrients-15-02115-t001:** Primer sequences used for RT-qPCR.

Gene	Acc. No	Primer Sequence (5’~3’)	Product Length (bp)
Mus *GPX4*	NM_001367995.1	F: GTACAGGAGCCAGGAGTAATC	134
		R: GGCTGGACTTTCATCCATTTC	
Mus *SLC7A11*	NM_011990.2	F: CTTCGATACAAACGCCCAGATA	119
		R: CTGAATGGGTCCGAGTAAGAG	
Mus *SOST*	NM_001379100.1	F: ATCTGCCTACTTGTGCACGC	179
		R: TCATAGGGATGGTGGGGAGG	
Mus *RANKL*	NM_001314054.1	F: CACAGCCCTCTCTCTTGAGC	188
		R: AGACTGTGACCCCCTTCCAT	
Mus *IL-6*	NM_001314054.1	F: CTGCAAGAGACTTCCATCCA	131
		R: AGTGGTATAGACAGGTCTGTTGG	
Mus *IL-1β*	NM_008361.4	F: GAAATGCCACCTTTGACAGTG	116
		R: TGGATGCTCTCATCAGGACAG	
Mus *TNF-α*	NM_013693.3	F: TGTCTCAGCCTCTCTCATT	153
		R: TGATCTGAGTGTGTGAGGGTCT	
Mus *IL-10*	NM_010548.2	F: ACTTGGGTTGCCAAGCCTTA	223
		R: GACACCTTGGTCTGGAGCTTA	
Mus *IL-4*	NM_021283.2	F: GGGTCTCAACCCCCAGCTA	200
		R: CGAGCTCACTCTCTGTGTGTT	
Mus *GAPDH*	NM_001379100.1	F: GTGAAGGTCGGTGTGAACGG	227
		R: TCCTGGAAGATGGTGATGG	

## Data Availability

All data are contained within the article and the accompanying Appendix A.

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
