# Peer review of "Resveratrol Alleviates Diabetic Periodontitis-Induced Alveolar Osteocyte Ferroptosis Possibly via Regulation of SLC7A11/GPX4"

_nutrients, 2023, doi:10.3390/nu15092115_

Round 1

Reviewer 1 Report

The experimental structure, to analyze whether the resveratrol could play a role on alveolar osteocyte ferroptosis induced by diabetic periodontitis is interesting and original.

However, there are several aspects that highlight critical issues relating to the results and interpretations provided by the authors and which require an in-depth review of the manuscript

Materials and Methods section.

Page 2, line 85: It is not clear how experimental diabetic periodontitis was induced in animals, please better detail in the paragraph, specifying step by step, for example: the diabetic form in animals was induced according to the following procedure: periodontitis was induced according to the following procedure. It cannot be assumed by a reader not well versed in the subject that the experimental procedure is clear.

Page 3, line 95, please report full name foe STZ.

Immunohistochemistry section

Please report the thickness of the sections, is it the same as used for eosin hematoxylin staining?

Page 4 line 143: according to which criterion was the concentration of the ferroptosis inhibitor ferrostatin-1 (1 μM); for the same reason, indicate the reason for choosing the reported concentrations for AGEs and LPS.

Page 5 line 174. Please indicate the concentrations used for Resveratrol treatments.

Enzyme-linked immunosorbent assay section:

Please report in the text the used measure units to express the concentrations of the tested cytokines.

Malon-dialdehyde assay: the same as above.

Transmission electron microscopy section:

Please, specify the magnifications used for the microscopic analyses.

Legends to figures 1 and 2: Significant difference between the groups: what groups? The authors may specify to which groups compared the p-values refer.

Moreover, the significance calculated on n=3 is not reliable, the authors should add further data to have a convincing statistical evaluation.

Figure 2 Panel A: a high significance has even been reported in figure 2, but from the trend of the graph it is rather unlikely that this is the trend. Authors should carefully check these values.

In figure 3 there are several cases in which the significance is not evident but which the authors certify that they have found it, for example for IL-4 and IL-10. Authors should double check the data.

The images of Capillary-based immunoassay analysis seem untrue. Please replace them with better quality ones to be acceptable.

Page 13 line 316: Can the authors confirm that the concentrations used in vivo can match those tested in vitro in evaluating the effects they found? Please report in this paragraph this information.

Discussion section

The authors in the discussion did not comment however that resveratrol at higher concentrations has an effect enhancing the release of inflammatory mediators. How can the authors argue for this action?

Furthermore, a fundamental aspect to accept the interpretation of the experimental results obtained in this study, the authors should report that the concentrations tested in vitro can reproduce what happens in a vital organism given that the concentrations of resveratrol do not always have beneficial effects, as shown by the same data experiments obtained by the authors.

Minor comments:

Please, write in vitro as well as in vivo in italic.

Author Response

Reply to REVIEWER#1

We thank the reviewer for the thorough revision of the manuscript and highly valuable comments. We have addressed all the comments and made the changes in the manuscript. The changes in the manuscript can be found in the red color text.

Comment 1. Page 2, line 85: It is not clear how experimental diabetic periodontitis was induced in animals, please better detail in the paragraph, specifying step by step, for example: the diabetic form in animals was induced according to the following procedure: periodontitis was induced according to the following procedure. It cannot be assumed by a reader not well versed in the subject that the experimental procedure is clear.

Reply: Thank you for the highly important comment. We rewrote the text as follows: “To establish diabetes, mice were fed a high-sugar diet (48 kcal % fat) from the 6-week to 14-week age. At 10 week-age, the mice were injected with Streptozotocin (STZ, 25 mg/kg dissolved in 0.1 mmol cold citric acid buffer, pH 4.6) for five consecutive days. Diabetic periodontitis was induced in diabetic mice by maxillary second molar ligation [30] and 100 μL P. gingivalis-derived lipopolysaccharides (LPS, 0.2 mg/mL)/twice a week injection from 11 to 13-week age. In the control group, 100 μL of normal saline/twice a week was injected. The fasting blood glucose level was measured in mice's tail vein blood using Glucometer (Beyotime Technology, China). In the resveratrol treatment group, 100 μL (6.5 μM) resveratrol was injected locally in the periodontitis site for 7 days from 13-week age. At 14-week age (8 weeks after the experiment) mice were sacrificed and the maxilla with surrounding periodontal soft tissues were fixed with paraformaldehyde for 2 days.”

Comment 2. Page 3, line 95, please report full name for  STZ.

Reply: We wrote the full name of STZ.

Comment 3. Immunohistochemistry section, Please report the thickness of the sections, is it the same as used for eosin hematoxylin staining?

Reply: We wrote the thickness of the tissue section in the immunohistochemistry section.

Comment 4. Page 4 line 143: according to which criterion was the concentration of the ferroptosis inhibitor ferrostatin-1 (1 μM); for the same reason, indicate the reason for choosing the reported concentrations for AGEs and LPS.

Reply: We provided the relevant references for the use of ferrostatin-1, AGEs, and LPS in the Methods section.

Comment 5. Page 5 line 174. Please indicate the concentrations used for Resveratrol treatments.

Reply: We wrote the concentrations of resveratrol used.

Comment 6. Enzyme-linked immunosorbent assay section: Please report in the text the used measure units to express the concentrations of the tested cytokines.

Reply: We added the relevant information.

Comment 7. Malon-dialdehyde assay: the same as above.

Reply: We added the relevant information.

Comment 8. Transmission electron microscopy section: Please, specify the magnifications used for the microscopic analyses.

Reply: We specify the magnification in the TEM images.

Comment 9. Legends to figures 1 and 2: Significant difference between the groups: what groups? The authors may specify to which groups compared the p-values refer.

Reply: We rewrote the figure legends.

Comment 10. Moreover, the significance calculated on n=3 is not reliable, the authors should add further data to have a convincing statistical evaluation.

Reply: We performed 3 independent in vitro experiments in triplicate.  We acknowledge that the sample size of n=3 is relatively small, which may limit the accurate interpretation of some results. However, this pilot data of in vitro and in vivo studies can illustrate the findings of the current study. We do agree that further studies should be carried out with increased animal numbers and observation time points.

Comment 11. Figure 2 Panel A: a high significance has even been reported in figure 2, but from the trend of the graph it is rather unlikely that this is the trend. Authors should carefully check these values.

Reply: We rechecked and confirmed that the statistical analysis of Figure 2 Panel A is correct.

Comment 12. In figure 3 there are several cases in which the significance is not evident but which the authors certify that they have found it, for example for IL-4 and IL-10. Authors should double check the data.

Reply: We rechecked the statistical analysis of Figure 3 (IL-10 and IL-4) graphs and found that there was no significant difference between AGEs+LPS and AGEs+LPS+Fer-1 groups. We corrected the figure accordingly.

Comment 13. The images of Capillary-based immunoassay analysis seem untrue. Please replace them with better quality ones to be acceptable.

Reply: The image of capillary-based immunoassay are different from the conventional western blot images. The original blots of Capillary-based immunoassay analysis are provided in supplementary Figure 3 as well as the end of this reply.

Comment 14. Page 13 line 316: Can the authors confirm that the concentrations used in vivo can match those tested in vitro in evaluating the effects they found? Please report in this paragraph this information.

Reply: We rewrote the text as follows: “These results further support the results of in vitro studies i.e., 6.5 μg/mL resveratrol mitigates the diabetic periodontitis condition-induced alveolar osteocyte ferroptosis.”

Comment 15. Discussion section: the authors in the discussion did not comment however that resveratrol at higher concentrations has an effect enhancing the release of inflammatory mediators. How can the authors argue for this action?

Comment 16. Furthermore, a fundamental aspect to accept the interpretation of the experimental results obtained in this study, the authors should report that the concentrations tested in vitro can reproduce what happens in a vital organism given that the concentrations of resveratrol do not always have beneficial effects, as shown by the same data experiments obtained by the authors.

Reply: To address comments 15 and 16, we added the following text in the discussion section “In previous studies regarding resveratrol treatment in the periodontitis animal model, resveratrol is administrated mainly via gavage or intraperitoneal injection [28, 82, 83].  In this study, we locally injected resveratrol 6.5 μg/mL multiple times in the periodontal region. Although the dose of systematically administrated resveratrol is comparatively high, systemically administrated resveratrol can be cleared from the body via urine, feces, and sweat minimizing the high-dose related adverse effects. However, higher dose-dependent side effects of resveratrol including nausea, flatulence, bowel motions, abdominal discomfort, loose stools, and diarrhea have been reported in various clinical trials [84]. We found a higher dose of resveratrol induces osteocyte inflammation in vitro and did not rescue the diabetic periodontitis-induced osteocyte cell death. Therefore, the dose-dependent effect of locally administrated resveratrol in periodontitis pathogenicity and osteocyte functions as well as local adverse effects should be thoroughly investigated in future studies.”

Comment 17. Minor comments: please, write in vitro as well as in vivo in italic.

Reply: We corrected this mistake.

Supplementary Figure 3: Original blot for Fig. 4E. Capillary-based immunoassay analysis of (A) GPX4, (B) SLC7A11, and (C) β-actin in osteocytes. Original blot for Fig. 6C. Capillary-based immunoassay analysis of (D) p-IKBα, (E) IKBα, (F) NF-κB p-p65, (G) NF-κB p65, and (H) β-actin in osteocytes.

Reviewer 2 Report

Comments resveratrol 

  • Provide more details about the procedure of counting the number of immunostaining positive cells. 

  • What was the reason to use only male mice and not female? 

  • In method´s paragraph the description of the ferroptosis markers used in immunostaining technique should be described 

  • Micro-computed tomography measured the bone level after elimination of soft tissues and histology measured the CEJ-ABC distance in soft tissues, the same specimen cannot be used. Describe how was obtained both samples. 

  • Supplementary fig 1 shows two groups, the results for the three groups should be provided. 

  • Also, fig 1 shows two groups and the results for the three groups should be provided.  

  • Fig 3 shows the differences between all the groups and control but differences between all the groups should be shown. 

  • In fig 3 how can it be explained that ferr-1 group does not show differences with control group? 

  • In vitro study used ferrostatin-1, why has it not been studied in mice model? 

  • According to method´s paragraph the intracellular Malon-dialdehyde assay (MDA) level was measured in cell lysate. Fig 4K shows the measurement of MDA in electron microscopy. Explain this issue. 

Author Response

Reply to REVIEWER#2

We thank the reviewer for the thorough revision of the manuscript and highly valuable comments. We have addressed all the comments and made the changes in the manuscript. The changes in the manuscript can be found in the red color text.

Comment 1. Provide more details about the procedure of counting the number of immunostaining positive cells.

Reply: We used the Allred scoring system as indicated below [1].

We wrote the relevant text in the Methods (immunohistochemistry) section as follows: “The immunostaining positive osteocytes (%) were counted using the Allred scoring system [33].”

Comment 2. What was the reason to use only male mice and not female?

Reply: To avoid the effect of hormonal fluctuation of female mice on bone cells’ function, male mice are commonly used during periodontitis studies. Therefore, we also used male mice.

Comment 3. In method´s paragraph the description of the ferroptosis markers used in immunostaining technique should be described 

Reply: We wrote the text in the Methods  (immunohistochemistry) section as follows: “GPX4, a master regulator of ferroptosis, interrupts lipid peroxidation by converting lipid hydroperoxides into non-toxic lipid alcohols [31]. SLC7A11 is a multi-pass transmembrane protein that mediates the cystine/glutamate antiporter activity in the system xc and regulates GPX4 expression and ferroptosis [32]. Therefore, in this study, we choose to perform the immunohistochemistry of GPX4 and SLC7A11.”

Comment 4. Micro-computed tomography measured the bone level after elimination of soft tissues and histology measured the CEJ-ABC distance in soft tissues, the same specimen cannot be used. Describe how was obtained both samples. 

Reply: Micro-CT followed by histology in the same bone tissue sample is well accepted and commonly used technique. Our research group and other groups have frequently used this protocol [2-5].

Comment 5. Supplementary fig 1 shows two groups, the results for the three groups should be provided. 

Comment 6. Also, fig 1 shows two groups and the results for the three groups should be provided.  

Reply to comments 5 and 6: The main aim of this study was to investigate the effect of diabetic periodontitis on alveolar osteocytes. Therefore, we designed two groups  (control and diabetic periodontitis group). We acknowledge that including only diabetes and only the periodontitis group could help to better understand the effect of co-morbidity, and this is the limitation of this study. We mentioned this issue in the discussion section As follows: “The lack of only periodontitis and only diabetes group is the limitation of this study. A future study including these groups will provide a better understanding of the role of resveratrol in alleviating diabetic periodontitis comorbidity-induced alveolar osteocyte ferroptosis.”

Comment 7. Fig 3 shows the differences between all the groups and control but differences between all the groups should be shown. 

Reply: We compared the statistical difference between the Control and Fer-1 groups, but there was no significant difference. We compared the AGEs+LPS group with the Control group and Fer-1 group. We updated this information in the figure legend. We also compared the AGEs+LPS+RSV group with the AGEs+LPS group.

Comment 8. In fig 3 how can it be explained that ferr-1 group does not show differences with control group?

Reply: We believe that in the control condition, GPX4 and SLC7A11 expressions are at their maximum level and ferroptosis does not occur. Therefore, the Fer-1 group does not show differences with the control group. A similar effect of Fer-1 is also reported in the literature [6].

Comment 9. In vitro study used ferrostatin-1, why has it not been studied in mice model?

Reply:  Since there was no significant effect of Fer-1 during in vitro study, we did not include Fer-1 in vivo study.  

Comment 10. According to method´s paragraph the intracellular Malon-dialdehyde assay (MDA) level was measured in cell lysate. Fig 4K shows the measurement of MDA in electron microscopy. Explain this issue. 

Reply: MDA was performed in cell lysate. To avoid this confusión, we rewrote the figure legend as follows: “(K) Analysis of MDA level in osteocyte cell lysate.”

References

  1. Fedchenko, N.; Reifenrath, J. Different approaches for interpretation and reporting of immunohistochemistry analysis results in the bone tissue - a review. Diagn Pathol 2014, 9, 221, doi:10.1186/s13000-014-0221-9.
  2. Wang, L.; Liang, D.; Huang, Y.; Chen, Y.; Yang, X.; Huang, Z.; Jiang, Y.; Su, H.; Wang, L.; Pathak, J.L., et al. SAP deficiency aggravates periodontitis possibly via C5a-C5aR signaling-mediated defective macrophage phagocytosis of Porphyromonas gingivalis. J Adv Res 2022, 10.1016/j.jare.2022.10.003, doi:10.1016/j.jare.2022.10.003.
  3. Li, Z.; Zheng, Z.; Pathak, J.L.; Li, H.; Wu, G.; Xu, S.; Wang, T.; Cheng, H.; Piao, Z.; Jaspers, R.T., et al. Leptin-deficient ob/ob mice exhibit periodontitis phenotype and altered oral microbiome. J Periodontal Res 2023, 58, 392-402, doi:10.1111/jre.13099.
  4. Zheng, Z.; Wu, L.; Li, Z.; Tang, R.; Li, H.; Huang, Y.; Wang, T.; Xu, S.; Cheng, H.; Ye, Z., et al. Mir155 regulates osteogenesis and bone mass phenotype via targeting S1pr1 gene. Elife 2023, 12, doi:10.7554/eLife.77742.
  5. Xiang, X.; Pathak, J.L.; Wu, W.; Li, J.; Huang, W.; Wu, Q.; Xin, M.; Wu, Y.; Huang, Y.; Ge, L., et al. Human serum-derived exosomes modulate macrophage inflammation to promote VCAM1-mediated angiogenesis and bone regeneration. J Cell Mol Med 2023, 27, 1131-1143, doi:10.1111/jcmm.17727.
  6. Liu, X.; Chen, C.; Han, D.; Zhou, W.; Cui, Y.; Tang, X.; Xiao, C.; Wang, Y.; Gao, Y. SLC7A11/GPX4 Inactivation-Mediated Ferroptosis Contributes to the Pathogenesis of Triptolide-Induced Cardiotoxicity. Oxid Med Cell Longev 2022, 2022, 3192607, doi:10.1155/2022/3192607.

Round 2

Reviewer 1 Report

No other comments

Reviewer 2 Report

The authors should provide a description of the answers to my comments, there is someone who has not attended.